# What Is Accounting for the Rapid Decline in Cigarette Sales in Japan?

**DOI:** 10.3390/ijerph17103570

**Published:** 2020-05-20

**Authors:** K. Michael Cummings, Georges J. Nahhas, David T. Sweanor

**Affiliations:** 1Department of Psychiatry and Behavioral Sciences, Medical University of South Carolina, Charleston, SC 29425, USA; elnahas@musc.edu; 2Faculty of Law, University of Ottawa, Ottawa, ON K1N 6N5, Canada; dsweanor@uottawa.ca

**Keywords:** cigarettes, marketing, policy, nicotine, prevention, epidemiology

## Abstract

This study describes how trends in the sale of cigarettes in Japan between 2011 and 2019 correspond to the sales of heated tobacco products (HTPs) that were introduced into the Japanese market in late 2015. Data used for this study come from the Tobacco Institute of Japan and Philip Morris International. The findings show that the accelerated decline in cigarette only sales in Japan since 2016 corresponds to the introduction and growth in the sales of HTPs.

## 1. Introduction

The substitution of non-combustion products has the potential to be a highly effective and non-coercive risk reduction strategy given the well-documented health risks of long-term smoking [1]. Heated tobacco products (HTPs) are devices that use heat processed tobacco rather than burn the tobacco directly in order to generate a nicotine aerosol for inhalation, which appears to have a lower risk profile compared to conventional tobacco cigarettes [2]. Japan has been a testing ground for HTPs [3,4,5,6]. IQOS (i.e., stands for “I Quit Ordinary Smoking”), marketed by Philip Morris International (PMI), was first introduced in 2014, followed in 2016 by the launches of Ploom TECH by Japan Tobacco International (JTI) and glo by British American Tobacco (BAT). According to market analyst reports, Japan has the most developed HTP market of all countries worldwide, accounting for 85% of HTP sales in 2018 [7]. 

This study describes how trends in the sale of cigarettes in Japan correspond to the sales of HTPs using data collected between 2011 and 2019. 

## 2. Methods

The limited data for the study comes from two sources: the Tobacco Institute of Japan (TIOJ) (https://www.tioj.or.jp/data/pdf/190424_02.pdf), and Philip Morris International, and was available for 2011 through 2019. The data from TIOJ is in Japanese, but an English translation copy is available upon request from the authors. Table 1 provides the raw data used in this study. Sales data are available for individual years, with sales measured in billion sticks. Trend analyses were performed in Joinpoint 4.7.0.0 to February 2019. Joinpoint regression models are used to describe continuous changes in trends using the grid-search method to fit the regression function with unknown joinpoints assuming constant variance and uncorrelated errors. More details about this free statistical tool can be found at https://surveillance.cancer.gov/joinpoint/ and in the paper by Kim et al. [8]. In brief, this is a software that fits the simplest joinpoint model to a set of data points. The program tests the statistical significance of no joinpoints (straight line) compared to one or more joinpoints. It displays a graph that includes the points, the fitted regression line, and the significant joinpoints (Appendix A).

## 3. Results 

Table 1 shows that between 2011 and 2019, overall cigarette sales declined by 38%, and total tobacco sales (i.e., combining cigarettes and HTPs) declined by 19%. Figure 1 plots the available data from Table 1 to display cigarettes sales, HTP sales, and combined cigarette and HTP sales. As illustrated, domestic cigarette sales in Japan appear to have declined at an accelerated pace since 2016 following the introduction of HTPs into the Japanese national marketplace. Using joinpoint analyses, overall cigarette and HTP sales had an annual percent change (APC) of −4.77 (*p* < 0.0001), between 2011 and 2019. Between 2016 to 2019, following the national marketing of HTPs (IQOS in 2016 and other HTPs in 2017), the APC was −6.69 (*p* = 0.0092). However, separating out cigarette sales from HTP sales reveals a different pattern. Between 2011 and 2015, the APC for cigarette sales was −3.10 (*p* = 0.1066); between 2016 and 2019, the APC was −16.38 (*p* = 0.0004), a difference of 13.28 (*p* = 0.0033) representing a five-fold increase compared to the pre-HTP period. 

## 4. Discussion

Between 2011 and 2015, cigarette sales in Japan were declining at a slow but steady pace. However, the pace of decline in cigarette sales accelerated beginning in 2016, corresponding to the introduction of HTPs into the marketplace. This finding is consistent with the conclusion of Stoklosa and colleagues [3] who examined data on sales of tobacco products from participating supermarkets and convenience stores in different regions of Japan between 2014 and 2018. The accelerated decline in cigarette sales in Japan after 2016 is rather remarkable since it appears to have happened independent of efforts made by public health groups that have largely opposed the marketing of HTPs [9]. Also, Japan does not have strong smoking control measures in place and prohibits the marketing of electronic nicotine delivery systems (ENDS), which have been associated with declining cigarette sales in the United States and England [10,11,12].

This study does not address the extent to which individual cigarette smokers are substituting HTPs for conventional cigarettes. A recent study suggests that most HTP users in Japan are also concurrently smoking cigarettes [13]. That said, these data do suggest that in Japan at least, the decline in cigarette sales has been accelerated by the introduction of HTPs. It is hard to know if the findings in Japan can be replicated globally, but reported sales trends in other markets where HTPs have been introduced show a similar inverse association between cigarette and HTP sales [14]. Given the hype associated with HTPs, manufacturers need to do more to share their marketing data with public health officials and investors so that individual-level cigarette substitution and harm reduction from smoking can be accessed. Given the history of the cigarette industry, public health groups have a right to be skeptical of any industry product claims, however assuming all tobacco/nicotine products as equivalently harmful is also counterproductive to public health goals as it only serves to protect the most lethal nicotine product—cigarettes. The evolving marketplace of potentially lower-risk nicotine products of which HTPs are just one category, combined with regulatory authority over tobacco products, represents a new opportunity to dramatically transform the cigarette business in ways that were never imagined when the war on tobacco was raging decades ago. However, this requires embracing risk-proportionate regulatory and taxation policies and providing consumers with accurate public messaging on product relative risks [15]. One can only imagine what might be accomplished if market forces were aligned with public health goals to reduce premature deaths caused by smoking. 

## 5. Conclusions

The accelerated decline in cigarette only sales in Japan since 2016 corresponds to the introduction and growth in the sales of HTPs.

## Figures and Tables

**Figure 1 ijerph-17-03570-f001:**
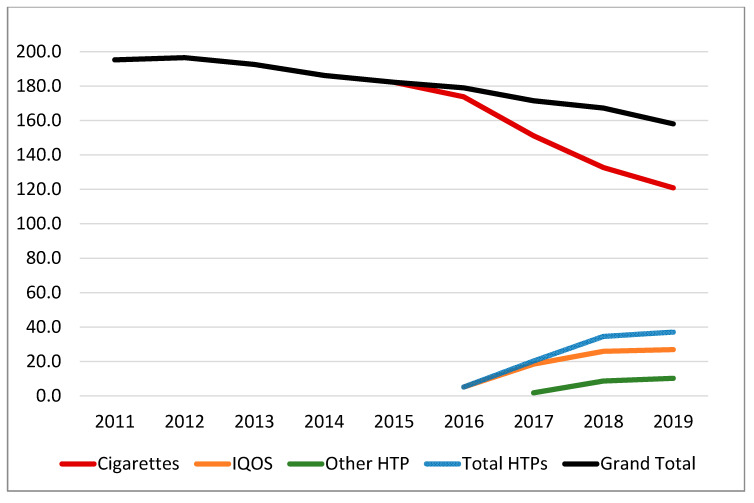
Sales of cigarettes, IQOS, and other HTPs (billion sticks).

**Table 1 ijerph-17-03570-t001:** Sales of tobacco products in Japan, 2011–2019 ^1^.

Calendar Year	Cigarettes		IQOS (I Quit Ordinary Cigarettes)		Other HTP		Total HTPs		Grand Total
	N	%	N	%	N	%	N	%	
2011	195.3	100.0							195.3
2012	196.6	100.0							196.6
2013	192.6	100.0							192.6
2014	186.2	100.0							186.2
2015	182.3	100.0							182.3
2016	173.8	97.1	5.1	2.9			5.1	2.9	179.0
2017	151.2	88.2	18.5	10.8	1.8	1.0	20.3	11.8	171.5
2018	132.7	79.3	25.9	15.5	8.7	5.2	34.6	20.7	167.3
2019	120.9	76.5	26.9	17.0	10.2	6.5	37.1	23.5	158.1

^1^ The data sources for the information included in Table 1 come from several sources. Conventional cigarette volume comes from the Tobacco Institute of Japan (TIOJ): converted to show the volume of sales in a calendar year in billion sticks. Annual cigarette volume prior to 2016 was obtained from PMI’s earnings reports (https://www.pmi.com/investor-relations/reports-filings), which itself is based on the TIOJ data. IQOS sales data comes from Philip Morris International’s (PMI’s) quarterly earnings reports and were calculated from the reported market share of heatsticks. The other heated tobacco product (HTP) volume is computed as the total market volume less heatstick volume less cigarette volume. We recognize that other HTPs such as Ploom TECH consumables pack consist of five tobacco capsules and one liquid cartridge. Japan Tobacco asserts that one pack of Ploom TECH consumables is equivalent to one pack of 20 combustible cigarette sticks. We used this conversion in the data presented in the table. The total HTP figures shown in the table are determined by adding heatstick volume with other HTP volume.

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
