# Peer review of "What Is Accounting for the Rapid Decline in Cigarette Sales in Japan?"

_ijerph, 2020, doi:10.3390/ijerph17103570_

Round 1

Reviewer 1 Report

This work investigated the cause of the rapid decline in cigarette sales in Japan. The results are interesting and helpful for the countries that is going to introduce HTP. However, it is not enough to conclude that the decline in cigarette sales since 2016 correspond to the introduction of HTP. I recommend that authors should analyze not only national sales but also some main local sales. It is also necessary to analyze cigarette sales with smoking rate by age. The manuscript can be accepted after more investigation.

P.S.
I recommend you to make sure if www.mdpi.com/xxx/s1 is available.

Author Response

Comments and Suggestions for Authors

This work investigated the cause of the rapid decline in cigarette sales in Japan. The results are interesting and helpful for the countries that is going to introduce HTP. However, it is not enough to conclude that the decline in cigarette sales since 2016 correspond to the introduction of HTP. I recommend that authors should analyze not only national sales but also some main local sales. It is also necessary to analyze cigarette sales with smoking rate by age. The manuscript can be accepted after more investigation.

Response:

We appreciate the comment and wish we had more data available to share. We do not have local data, or prevalence data which would allow us to look at trends by age, gender and other factors. We agree with the reviewer such data are desirable. We cited the paper by Sustanto et al., (see reference 13) which is one of the few population surveys available on smoking and HTP use, although this study is based on a single cross-sectional survey. The simple purpose of our study was to share the data we were able to acquire showing the trends in smoking and Heated Tobacco Product (HTPs) sales. These are unique data since Japan Tobacco up until 2018 had included HTPs as part of overall cigarette sales in their publicly released data. In this revision we have added data for 2019 which further demonstrated the inverse relationship between cigarette sales and HTP sales. To address the reviewers concerns about the limitation of our study we have added the following paragraph to the discussion:

This study does not address the extent to which individual cigarette smokers are substituting HTPs for conventional cigarettes. A recent study suggests that most HTP users in Japan are also concurrently smoking cigarettes [13]. That said, these data do suggest that in Japan at least, the decline in cigarettes sales has been accelerated by the introduction of HTPs.   It is hard to know if the findings in Japan can be replicated globally but reported sales trends in other markets where HTPs have been introduced show a similar inverse association between cigarette and HTP sales [14]. Given the hype associated with HTPs, manufacturers need to do more to share their marketing data with public health officials and investors so that individual level cigarette substitution and harm reduction from smoking can accessed. Given the history of the cigarette industry, public health groups have a right to be skeptical of any industry product claims, however assuming all tobacco/nicotine products as equivalently harmful is also counterproductive to public health goals as it only serves to protect the most lethal nicotine product – cigarettes.   The evolving marketplace of potentially lower risk nicotine products of which HTPs are just one category, combined with regulatory authority over tobacco products represents a new opportunity to dramatically transform the cigarette business in ways that were never imagined when the war on tobacco was raging decades ago. However, this requires embracing risk-proportionate regulation, and taxation policies and providing consumers with accurate public messaging on product relative risks [15]. One can only imagine what might be accomplished if market forces were aligned with public health goals to reduce premature deaths caused by smoking.  

Reviewer 2 Report

This brief report (letter) purports to show two things: 1) that the decline in cigarette sales over a short period of time in Japan is the result of increased heat not burn (HNB) product sales, using data supplied by the tobacco industry on sales, and 2) that this decline could and should be replicated in other countries around the world.

Unfortunately, this data is far too simple to adequately make either hypothesis supported fully in this report. For instance, they used 2011 -2106 to show declines in cigarette sales, what if a shorter period was used? Longer period? Sales were clearly decreasing, and if you use a shorter period and eliminate one year with little change, there is not much to report as a different rate of decline.  Also, there is no way to prove in this model whether the increase in sales of HNB product use was by adults or adolescents not using cigarettes (though less likely), or whether cigarette users simply cut back on their cigarette use while using HNB but remained dual users to some extent.  These scenarios would alter both of the 'findings' of the authors. This is not a detailed analysis of what efforts have been occurring in Japan related to all tobacco product use.  Finally, the statements that more would be accomplished if public health groups would encourage smokers to abandon cigarettes in favor of HNB products is an opinion based on belief this a large public health gain, data that this paper does not provide. That opinion, while plausible, is also controversial, as WHO and many other respected health authorities have not advocated this opinion and simultaneously voiced legitimate concerns not represented.

Author Response

Comments and Suggestions for Authors

This brief report (letter) purports to show two things: 1) that the decline in cigarette sales over a short period of time in Japan is the result of increased heat not burn (HNB) product sales, using data supplied by the tobacco industry on sales, and 2) that this decline could and should be replicated in other countries around the world.

Unfortunately, this data is far too simple to adequately make either hypothesis supported fully in this report. For instance, they used 2011 -2106 to show declines in cigarette sales, what if a shorter period was used? Longer period? Sales were clearly decreasing, and if you use a shorter period and eliminate one year with little change, there is not much to report as a different rate of decline.  Also, there is no way to prove in this model whether the increase in sales of HNB product use was by adults or adolescents not using cigarettes (though less likely), or whether cigarette users simply cut back on their cigarette use while using HNB but remained dual users to some extent.  These scenarios would alter both of the 'findings' of the authors. This is not a detailed analysis of what efforts have been occurring in Japan related to all tobacco product use.  Finally, the statements that more would be accomplished if public health groups would encourage smokers to abandon cigarettes in favor of HNB products is an opinion based on belief this a large public health gain, data that this paper does not provide. That opinion, while plausible, is also controversial, as WHO and many other respected health authorities have not advocated this opinion and simultaneously voiced legitimate concerns not represented.

Response:

We agree with the reviewer that the data we have are limited. We were able to get sales data for another year (i.e., 2019) and have added these data to the revised paper. The inverse association between cigarette sales heated tobacco product (HTP) sales remains.  

We do recognize the limitations of study and have highlighted these limitations in our discussion. We’ve also dropped the sentence mentioned by reviewer. The final paragraph of the discussion now reads as follows:

This study does not address the extent to which individual cigarette smokers are substituting HTPs for conventional cigarettes. A recent study suggests that most HTP users in Japan are also concurrently smoking cigarettes [13]. That said, these data do suggest that in Japan at least, the decline in cigarettes sales has been accelerated by the introduction of HTPs. It is hard to know if the findings in Japan can be replicated globally but reported sales trends in other markets where HTPs have been introduced show a similar inverse association between cigarette and HTP sales [14]. Given the hype associated with HTPs, manufacturers need to do more to share their marketing data with public health officials and investors so that individual level cigarette substitution and harm reduction from smoking can accessed. Given the history of the cigarette industry, public health groups have a right to be skeptical of any industry product claims, however assuming all tobacco/nicotine products as equivalently harmful is also counterproductive to public health goals as it only serves to protect the most lethal nicotine product – cigarettes.   The evolving marketplace of potentially lower risk nicotine products of which HTPs are just one category, combined with regulatory authority over tobacco products represents a new opportunity to dramatically transform the cigarette business in ways that were never imagined when the war on tobacco was raging decades ago. However, this requires embracing risk-proportionate regulation, and taxation policies and providing consumers with accurate public messaging on product relative risks [15]. One can only imagine what might be accomplished if market forces were aligned with public health goals to reduce premature deaths caused by smoking.  

Reviewer 3 Report

IJERPH – Manuscript ID #764973

Title: What is accounting for the rapid decline in cigarette sales in Japan?

Authors: K. Michael Cummings * , Georges J. Nahhas , David T. Sweanor

Summary: The authors present compelling evidence of population level tobacco harm reduction after introduction of noncombustible tobacco products. This study describes how trends in the sale of cigarettes in Japan correspond to the sales of heated tobacco products (HTPs) that were introduced into the entire Japanese market in late 2015. The authors use data from the Tobacco Institute of Japan and Phillip Morris International Quarterly Earnings Reports. The findings show that accelerated decline in cigarette only sales in Japan since 2016 corresponds to the introduction and growth in the sales of HTPs.

Reviewers Comments:

This reviewer has the following comments:

1)      The authors cite the Tobacco Institute of Japan as one of the sources for the data, however the corresponding table shown in the supplementary file is in Japanese.  Please revise the table with the content included in English language.

2)      The  table 1 indicates observations for other HTPs.  Please provide the source of data for the other HTPs.

3)      The data presented in Table 1 indicates that the total tobacco consumption is declining, despite the increase in prevalence of HTP.  This is an important observation that should be highlighted by the authors.

4)      On Page 2, lines 53-55 the authors state –

“ Between 2011 and 2016, the APC for cigarette sales was -4.66 (p= 53 0.0091); between 2016 and 2018, the APC was -22.25 (p=0.0076), a difference of 17.59 (p=.0156), a 5-fold increase compared to the pre-HTP period.” 

                Please revise the second part of the question.  The difference in cigarette sales is an incremental 5-fold incremental decrease (not increase) compared to the pre-HTP period.  Although mathematically the difference between -4.66 and -22.25 is a positive number, the key takeaway of this analysis is that there is a sharp decline in the APC between 2016 and 2018 that is far more rapid than during the period between 2011-2016.  Please consider revising this statement for better clarity.

5)      The authors must include in the discussion section a description of limitations.  For example one of the limitations could be that these observations  do not necessarily reflect a cause and effect relationships.  Additional confirmation will be necessary to determine whether these trends continue over a longer duration.

6)      The authors should consider including a brief description of the overall impact of these findings.  Perhaps the lack of anti-harm reduction messaging in Japan and erosion of consumer perceptions regarding HTPs may have interfered with the adoption of the novel non-combustible products. Additional research is needed regarding risk perceptions of the consumers.

Author Response

Comments and Suggestions for Authors

IJERPH – Manuscript ID #764973

Title: What is accounting for the rapid decline in cigarette sales in Japan?

Authors: K. Michael Cummings * , Georges J. Nahhas , David T. Sweanor

Summary: The authors present compelling evidence of population level tobacco harm reduction after introduction of noncombustible tobacco products. This study describes how trends in the sale of cigarettes in Japan correspond to the sales of heated tobacco products (HTPs) that were introduced into the entire Japanese market in late 2015. The authors use data from the Tobacco Institute of Japan and Phillip Morris International Quarterly Earnings Reports. The findings show that accelerated decline in cigarette only sales in Japan since 2016 corresponds to the introduction and growth in the sales of HTPs.

Reviewers Comments:

This reviewer has the following comments:

1)      The authors cite the Tobacco Institute of Japan as one of the sources for the data, however the corresponding table shown in the supplementary file is in Japanese.  Please revise the table with the content included in English language.

2)      The  table 1 indicates observations for other HTPs.  Please provide the source of data for the other HTPs.

3)      The data presented in Table 1 indicates that the total tobacco consumption is declining, despite the increase in prevalence of HTP.  This is an important observation that should be highlighted by the authors.

4)      On Page 2, lines 53-55 the authors state –

“ Between 2011 and 2016, the APC for cigarette sales was -4.66 (p= 53 0.0091); between 2016 and 2018, the APC was -22.25 (p=0.0076), a difference of 17.59 (p=.0156), a 5-fold increase compared to the pre-HTP period.” 

                Please revise the second part of the question.  The difference in cigarette sales is an incremental 5-fold incremental decrease (not increase) compared to the pre-HTP period.  Although mathematically the difference between -4.66 and -22.25 is a positive number, the key takeaway of this analysis is that there is a sharp decline in the APC between 2016 and 2018 that is far more rapid than during the period between 2011-2016.  Please consider revising this statement for better clarity.

5)      The authors must include in the discussion section a description of limitations.  For example one of the limitations could be that these observations  do not necessarily reflect a cause and effect relationships.  Additional confirmation will be necessary to determine whether these trends continue over a longer duration.

6)      The authors should consider including a brief description of the overall impact of these findings.  Perhaps the lack of anti-harm reduction messaging in Japan and erosion of consumer perceptions regarding HTPs may have interfered with the adoption of the novel non-combustible products. Additional research is needed regarding risk perceptions of the consumers

Response:

Response to comments 1. Below is the table from Japan Tobacco Institute translated into English. We have added a sentence in the paper informing readers they can obtain an English Language transition from the authors. See below:

The data from TIOJ is in Japanese, but an English translation copy is available upon request from the authors.

Below is the table translated into English.

Japan Tobacco Annual Sales

List of sales results (Quantity/Price) by year

(Unit: 100 million, 100 million yen,%)

Year

Sales Quantity

Sales Price

Japanese Calendar

AD

Total

(Previous Year)

Total

(Previous Year)

Heisei 2

1990

3,220

(102.6)

35,951

(103.3)

Hesisi 3

1991

3,283

(102.0)

36,965

(102.8)

Hesisi 4

1992

3,289

(100.2)

37,216

(100.7)

Hesisi 5

1993

3,326

(101.1)

37,817

(101.6)

Hesisi 6

1994

3,344

(100.5)

38,183

(101.0)

Hesisi 7

1995

3,347

(100.1)

38,327

(100.4)

Hesisi 8

1996

3,483

(104.1)

39,992

(104.3)

Hesisi 9

1997

3,280

(94.2)

38,971

(97.4)

Hesisi 10

1998

3,366

(102.6)

40,899

(104.9)

Heisei 11

1999

3,322

(98.7)

42,600

(104.2)

Hesisi 12

2000

3,245

(97.7)

41,681

(97.8)

Hesisi 13

2001

3,193

(98.4)

41,037

(98.5)

Hesisi 14

2002

3,126

(97.9)

40,187

(97.9)

Hesisi 15

2003

2,994

(95.8)

40,660

(101.2)

Hesisi 16

2004

2,926

(97.7)

40,682

(100.1)

Hesisi 17

2005

2,852

(97.5)

39,694

(97.6)

Hesisi 18

2006

2,700

(94.7)

39,820

(100.3)

Hesisi 19

2007

2,585

(95.7)

39,131

(98.3)

Hesisi 20

2008

2,458

(95.1)

37,270

(95.2)

Hesisi 21

2009

2,339

(95.1)

35,460

(95.1)

Hesisi 22

2010

2,102

(89.9)

36,163

(102.0)

Hesisi 23

2011

1,975

(94.0)

41,080

(113.6)

Hesisi 24

2012

1,951

(98.8)

40,465

(98.5)

Hesisi 25

2013

1,969

(100.9)

40,744

(100.7)

Hesisi 26

2014

1,793

(91.0)

38,418

(94.3)

Hesisi 27

2015

1,833

(102.2)

39,227

(102.1)

Hesisi 28

2016

1,680

(91.7)

36,377

(92.7)

Hesisi 29

2017

1,455

(86.6)

31,655

(87.0)

Hesisi 30

2018

1,300

(89.3)

29,368

(92.8)

*Because the number is rounded off to the nearest unit, it may be incorrect

Response to comment 2. We have added a more detailed footnote to table 1 providing information on the various sources relied upon for the data contained in table 1.

The data sources for the information included in Table 1 come from several sources.

Conventional cigarette volume comes from the Tobacco Institute of Japan (TIOJ): converted to show the volume of sales in a calendar year in billion sticks. Annual cigarette volume prior to 2016 was obtained from PMI’s earnings reports (https://www.pmi.com/investorrelations/reports-filings), which itself is based on the TIOJ data. IQOS sales data comes from PMI‘s quarterly earnings reports and were calculated from the reported market share of Heat sticks. The other HTP volume is computed as the total market volume less heat stick volume less cigarette volume. We recognize that other HTPs such as Ploom Tech consumables pack consist of five tobacco capsules and one liquid cartridge. Japan Tobacco asserts that one pack of Ploom Tech consumables is equivalent to one pack of 20 combustible cigarette sticks. We used this conversion in the data presented in the table. The total HTP figures shown in the table are determined by adding heat stick volume with other HTP volume.

Response to comment 3. We agree with the reviewer’s comment about declining total tobacco sales and have added this to the results as follows:

Table 1 shows that between 2011 and 2019, overall cigarette sales declined by 38%, and total tobacco sales (i.e., combining cigarettes and HTPs) declined by 19%.

Response to comment 4. The statement was awkward as we were noting the increase annual percent change. However, because we have added data on sales for 2019, all analyses were updated and the results section re-written. The results now reads as follows:

Using joinpoint analyses, between 2011 and 2019 overall cigarette and HTP sales had an annual percent change (APC) of -4.77 (p<.0001). Between 2016 to 2019 following the national marketing of HTPs (IQOS in 2016 and other HTPs in 2017) the APC was   -6.69 (p=.0092). However, separating out cigarette sales from HTP sales reveals a different pattern. Between 2011 and 2015, the APC for cigarette sales was -3.10 (p=.1066); between 2016 and 2019, the APC was -16.38 (p=0.0004), a difference of 13.28 (p=.0033), a 5-fold increase compared to the pre-HTP period.

Response to comment 5. We acknowledge this is a descriptive study and have stated this in the abstract and the in the introduction to the methods section of the paper. We also accept the limitations of the study of this descriptive study and have added the following statement to the discussion: 

This study does not address the extent to which individual cigarette smokers are substituting HTPs for conventional cigarettes. A recent study suggests that most HTP users in Japan are also concurrently smoking cigarettes [13]. That said, these data do suggest that in Japan at least, the decline in cigarettes sales has been accelerated by the introduction of HTPs.   It is hard to know if the findings in Japan can be replicated globally but reported sales trends in other markets where HTPs have been introduced show a similar inverse association between cigarette and HTP sales [14]. Given the hype associated with HTPs, manufacturers need to do more to share their marketing data with public health officials and investors so that individual level cigarette substitution and harm reduction from smoking can accessed. Given the history of the cigarette industry, public health groups have a right to be skeptical of any industry product claims, however assuming all tobacco/nicotine products as equivalently harmful is also counterproductive to public health goals as it only serves to protect the most lethal nicotine product – cigarettes.   The evolving marketplace of potentially lower risk nicotine products of which HTPs are just one category, combined with regulatory authority over tobacco products represents a new opportunity to dramatically transform the cigarette business in ways that were never imagined when the war on tobacco was raging decades ago. However, this requires embracing risk-proportionate regulation, and taxation policies and providing consumers with accurate public messaging on product relative risks [15]. One can only imagine what might be accomplished if market forces were aligned with public health goals to reduce premature deaths caused by smoking.  

Reviewer 4 Report

Dear Authors

I suggest transforming this into a short communication instead of an article.

Every word in Latin like et al. must be in italic once it is Latin.

Author Response

Comments and Suggestions for Authors

Dear Authors

I suggest transforming this into a short communication instead of an article.

Every word in Latin like et al. must be in italic once it is Latin.

Response:

We agree that this is a short communication, not an article.

We have used italic for et al. in references

Round 2

Reviewer 1 Report

This work investigated the cause of the rapid decline in cigarette sales in Japan. The manuscript can be accepted because the results are interesting and helpful for the countries that is going to introduce HTP.